# Transfering Knowledge into Efficient Tiny Models for Object Detection with Dual Prompt Distillation

## Abstract

Knowledge Distillation (KD) has demonstrated significant benefits for learning compact models for object detection. Most current work focuses on general distillation settings, where student models are relatively large and learnable, then compete with the distillation performance. However, due to the model scale and inference speed, these models are seldom deployed in real-world applications. In this paper, we dive into a challenging but more applicable setting: *how to distill rich teacher knowledge into tiny, faster models for object detection?* We first show that simply applying previous KD strategies under such settings cannot achieve satisfying results, due to the extremely large model capacity gap between the teacher-student pairs. To this end, we propose a simple prompt-based object detection distillation framework, namely DualPromptKD, which aims to improve knowledge transfer efficiency from both teacher and student perspectives. Specifically, by distilling teacher representations into compact external prompts, we enable the student model to fully leverage proficient teacher knowledge even at inference time. In terms of the limited learning ability of the student model, we introduce lightweight internal prompts tailored to bolster the feature imitation capability for the target model. Extensive experimental results on the COCO benchmarks validate the effectiveness and generalization of our approach, including different image backbones and detector types. Notably, our DualPromptKD surpasses the previous best distillation strategies by more than 2.0 mAP under various experimental settings. The code will be available.

## 1 Introduction

The field of object detection has made remarkable advancements with the emergence of deep learning models (Cai & Vasconcelos, 2019; He et al., 2017; Tian et al., 2019; Li et al., 2020). However, the practical deployment of large and computationally intensive models in real-world applications poses significant challenges in terms of model size, inference speed, and resource constraints. Knowledge Distillation (KD) (Hinton et al., 2015; Chen et al., 2017; Chang et al., 2023; Cao et al., 2022), transferring knowledge from a well-performing teacher model to the target student model, has emerged as a promising technique to address these challenges. Current research primarily focuses on extracting scenarios where teacher and student models have comparable sizes (Huang et al., 2022a; Cho & Hariharan, 2019; Mirzadeh et al., 2020; Son et al., 2021; Cao et al., 2023). For example, DIST (Huang et al., 2022a) replaces the traditional KL divergence with a correlation-based loss function to better extract knowledge from a strong teacher model; MTPD (Cao et al., 2023) constructs a curriculum of teacher models to progressively transfer knowledge from complex teacher models to student model, effectively bridging the capacity gap and significantly enhancing the student's performance on object detection tasks. However, distilling knowledge into much smaller and faster models, which receives more attention in practical scenarios, is seldom discussed in previous studies.

In this paper, we delve into the problem of distilling rich teacher knowledge into efficient small models for object detection, considering the substantial model capacity gap. As shown in the Fig. 1, we attempt several state-of-the-art KD algorithms (Yang et al., 2022a; Cao et al., 2022; Huang et al., 2022a;b) to distill the GFL (Li et al., 2020) with GhostNet (Han et al., 2020), while they only achieve limited improvement. Among them, MasKD (Huang et al., 2022b) exhibits significantly

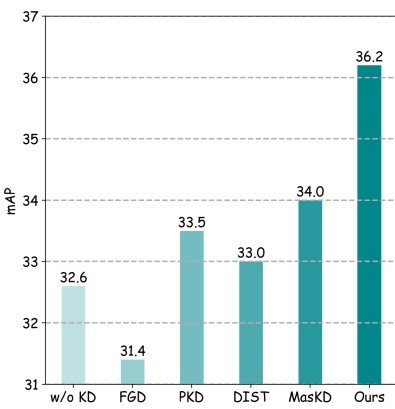

Figure 1: Comparing the performance of several current distillation methods with our approach on the COCO validation subset, where GFL-Res101 and GFL-GhostNet are utilized as the teacher and student models, respectively. The first column represents the baseline without distillation.

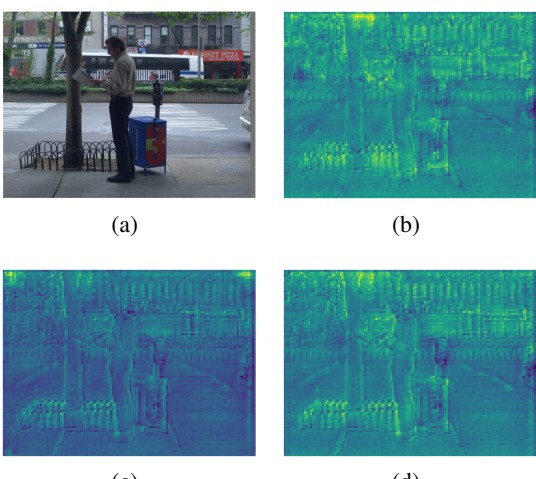

Figure 2: Visualization of the feature from the first layer of FPN outputs. **Teacher**:GFL-Res101. **Student**: GFL-GhostNet. (a) Original image. (b) Feature of teacher. (c) Feature of student. (d) Feature of student distilled with DualPromptKD.

smaller performance improvement compared to its ResNet-50 student model, and FGD (Yang et al., 2022a) even harms the performance of the model with extremely low parameter count. Besides, we further demonstrate that the large model capacity gap manifests as significant differences in the feature distributions of the teacher and student models, as shown in Fig. 2, primarily attributable to the varying feature extraction capabilities of their image backbones. Based on this, we have the following conclusions:

• When there is a significant disparity in parameter quantities between models, directly applying distillation can be suboptimal and even degrade performance due to the existence of the divergence in feature representations.

• Previous approaches have primarily focused on ResNet-50 style distillation on the detector, lacking effective supervision for the shallow layers of the backbone network.

• Large models, with their parameter redundancy, can effectively capture the relationship between the pre-training task and the detection task, facilitating transfer learning. However, efficient backbones, constrained by network capacity, often exhibit poor performance even on in-domain training tasks. Consequently, their performance saturates relative to network size, necessitating additional parameter quantities.

To address these issues, we propose DualPromptKD, an efficient knowledge distillation framework specialized for tiny detectors. Alongside the conventional feature distillation for FPN features, DualPromptKD incorporates two additional components: During the training process, we employ a set of *external prompts* to adaptively extract important representation characteristics from the teacher backbone using attention mechanisms, which are updated in a momentum-based manner. During the inference process, these extracted prompts are attached to the student backbone, enabling the student model to benefit from the comprehensive knowledge provided by the teacher. The attention mechanism, acting as a soft association, helps mitigate the domain gap between the teacher and student model representations. Additionally, we introduce lightweight *internal prompts* to guide the LoRA (Hu et al., 2021; Aleem et al., 2024) as adapters using dynamically generated masks, enhancing the feature extraction capability of the student model. The learnable prompts match student features through the hard association of dot products and the generated soft mask can highlight important areas while suppressing noisy areas. Internal prompts are only coupled with the student model, thereby preventing the drawback of blindly injecting knowledge from the teacher model.

We extensively evaluate our approach on the COCO benchmarks (Lin et al., 2014), considering various backbones and detector types. The experimental results validate the effectiveness and generalization of DualPromptKD, surpassing the previous state-of-the-art distillation strategies by more than 2.0 mAP under diverse experimental settings, demonstrating appealing robustness and practicality.

In summary, our proposed DualPromptKD framework offers an efficient solution for knowledge distillation on object detection, bridging the gap between large teacher models and compact student models. We hope it can provide a promising approach for the practical deployment of highly performant yet computationally efficient object detection models.

## 2 RELATED WORKS

### 2.1 KNOWLEDGE DISTILLATION FOR DETECTION

Knowledge Distillation (KD) (Romero et al., 2014; Huang & Wang, 2017; Liu et al., 2019; Wang et al., 2019; Zhang & Ma, 2020), is a kind of model compression and acceleration approach aiming at transferring knowledge from a teacher model to a student model. It was first proposed by Hinton (Hinton et al., 2015), using the output as soft labels to transfer the dark knowledge from a large teacher network to a small student network for the classification task. Recently, some works have successfully applied knowledge distillation to detectors (Cao et al., 2022; Yang et al., 2022a; Chang et al., 2023; Lao et al., 2023). Chen et al. (2017) first calculated the distillation loss on the detector's neck and head. The key to distillation for object detection is where to distill, due to the extreme imbalance between foreground and background. PKD (Cao et al., 2022) proposes imitating features with Pearson Correlation Coefficient to focus on the relational information from the teacher and relax constraints on the magnitude of the features. FGD (Yang et al., 2022a) proposes focal distillation which forces the student to learn the teacher's crucial parts and global distillation which compensates for missing global information. MGD (Yang et al., 2022b) first proposes masking out the feature maps in the knowledge distillation branch and using a generator to restore the teacher feature. And MasKD (Huang et al., 2022b) distills the valuable information in the features and ignores the noisy regions by learning to identify receptive regions that contribute to the task precision.

### 2.2 PROMPT LEARNING

Prompt-based learning approaches have been extensively studied in NLP (Liu et al., 2023; Schick & Schütze, 2020; Shin et al., 2020). The pioneer language model as GPT-3 (Brown et al., 2020) has shown its great few-shot or zero-shot potential across various tasks. The core of prompt-based learning is to modify the input sample into a prompted version and embed the expected output information as an unfilled slot inside the prompt. Prompting has also been applied to vision models recently. CLIP (Radford et al., 2021) introduces the prompt-based learning approach into the image recognition task by embedding the textual labels of the to-be-recognized objects into descriptive texts, and the classification procedure can be transformed into a video-text matching problem. CoOp (Zhou et al., 2022) utilizes learnable tokens as textual prompts and gains a promotion on few-shot image classification. There have also been initial approaches that attempt to prompt with images. CPT (Yao et al., 2024) converts visual grounding into a fill-in-the-blank problem by creating visual prompts with colored blocks and color-based textual prompts. Visual prompt tuning (Jia et al., 2022) proposes visual prompts specific to Vision Transformers (Dosovitskiy et al., 2020), using deep prompt tuning (Li & Liang, 2021) by prepending a set of tunable parameters to each Transformer encoder layer.

### 2.3 EFFICIENT TINY MODELS

In order to deploy on mobile devices for real-world applications, many light-weight CNN models with reduced parameter amounts and limited computational burdens are proposed (Howard et al., 2017; Chollet, 2017; Han et al., 2020; Chen et al., 2023). MobileNetV1 (Howard et al., 2017) and Xception (Chollet, 2017) propose the depth-wise separable convolution to decouple the regular convolution into depth-wise convolution and point-wise convolution, which alleviates a large amount of computation and parameters and has been a widely-adopted design element for modern efficient

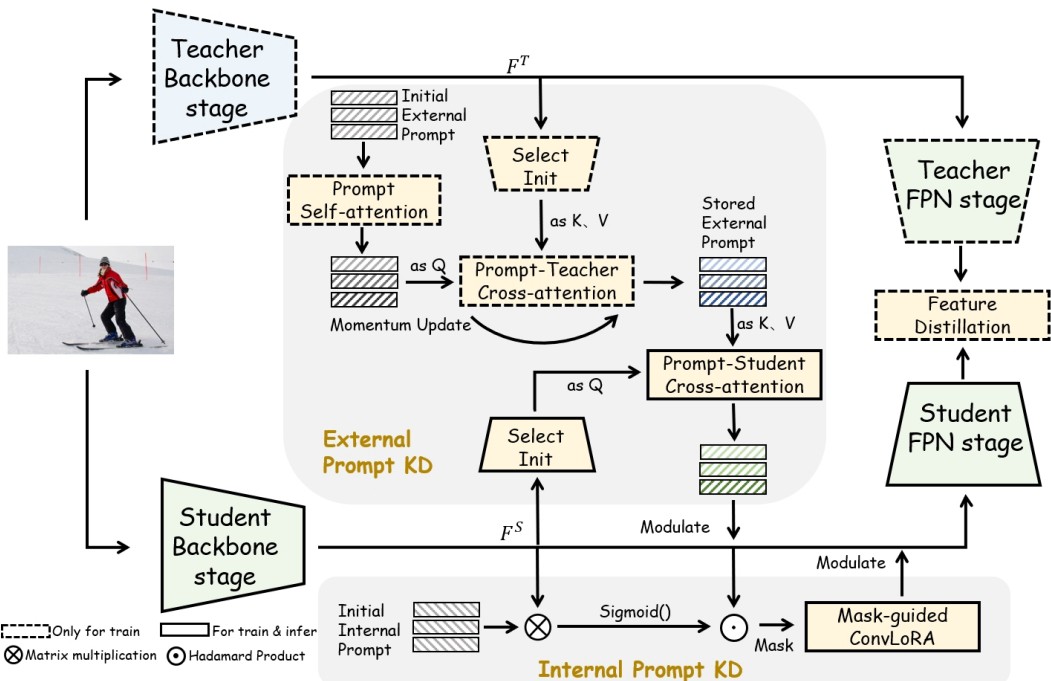

Figure 3: **Overview of DualPromptKD**. We first perform our feature distillation on the feature pyramid and then introduce two additional prompt KD. Among them, external prompt KD establishes a soft association to inject knowledge into the student from the teacher model, while internal prompt KD utilizes the hard connection to store student-relevant knowledge and performs enhancement.

CNN models. MobileNetV2 (Sandler et al., 2018) introduces the inverted residual block. MobileNetV3 (Howard et al., 2019) enhances MobileNetV2 with squeeze-and-excitation module and neural architecture search. GhostNet (Han et al., 2020) utilize a few small filters to generate more feature maps from the original convolutional layer, with an extremely efficient architecture and high performance. FasterNet (Chen et al., 2023) raises partial convolution to conduct regular convolution on part of the channels, which not only reduces the number of floating point operations required but also increases the processing speed per second.

However, when these lightweight networks are used as the backbone for the student model during distillation, the distillation effect is often unsatisfactory. This is due to the significant differences between the models, making it challenging for the student model to acquire effective knowledge. In this paper, we propose a novel prompt-based object detection distillation method that focuses more on supplementing feature information for small models, thereby achieving better performance.

## 3 METHOD

The objective of our work is to present an extension method for knowledge distillation that can be applied in extreme situations where there are significant differences between the two models. Fig. 3 illustrates the three pipelines that make up our proposed methodology: Feature Distillation, Internal Prompt Distillation, and External Prompt Distillation.

### 3.1 FEATURE DISTILLATION

The feature distillation process follows the current paradigm of feature-level distillation for detectors (Cao et al., 2022; Yang et al., 2022b). It utilises models with a large-parameter backbone as the teachers and models with a small-parameter backbone as the students. This is because most detectors utilise FPN (Lin et al., 2017a) to aggregate multi-scale information. Consequently, the most typical

manner is to transfer knowledge from the teacher to the student through the feature map after the neck. Feature distillation increases the similarity of features between the two models pixel-wisely, allowing students to obtain additional supervision with richer information. Formally, the distillation of the features can be expressed as follows:

$$\mathcal{L}_{feat} = \frac{1}{CHW} \sum \mathcal{M} \left( F^T - f(F^S) \right)^2, \tag{1}$$

where $F^T \in \mathbb{R}^{H \times W \times C^T}$ and $F^S \in \mathbb{R}^{H \times W \times C^S}$ denote the feature of the teacher and student, respectively. $H$, $W$ denote the height and width of the feature map and C is the channel. $f$ is a projection layer to adapt the channel of $F^S$ to the same as $F^T$. The mask $\mathcal{M}$ is a filter, and recent methods often customise different $\mathcal{M}$ to select meaningful regions for KD. In this section, we adopt the strategy employed in PKD (Cao et al., 2022), whereby the mask is filled with a scalar value of 1 rather than being delicately designed. We request that the normalized student features, denoted by $\hat{F^S}$, imitate the normalized features of the teacher, denoted by $\hat{F^T}$, as per Eq. 2. We place the task of selecting important features in the upstream backbone, prior to the FPN, as described in the next section.

$$\mathcal{L}_{feat} = \frac{1}{CHW} \sum \left( \hat{F^T} - f(\hat{F^S}) \right)^2, \tag{2}$$

## 3.2 PROMPT DISTILLATION

Although current feature-level distillations have achieved superior performance, traditional paradigms are inherently limited. To illustrate, distilling features after the neck can facilitate the propagation of supervised signals throughout the entire feature extraction module. However, the gradient of KD signals is prone to disappear in shallow stages, making it difficult to optimize effectively. Furthermore, when the discrepancy in the number of parameters between the teacher and student models is further amplified, constrained by the capacity of the model, relying solely on the student is insufficient to accurately predict the teacher's output. To address these limitations, we propose the introduction of the Prompt Distillation technique, which utilizes additional inserted prompts as a storage medium for knowledge, effectively bridging the performance gap between the teacher and student models at minimal additional cost.

### 3.2.1 EXTERNAL PROMPT DISTILLATION

The teacher model exhibits superior feature extraction capabilities due to its intricate and precise structure, which enables the effective enhancement of the information in the foreground region and the suppression of noise in the background region. The utilisation of only the predicted features of the teacher model as supervision is a suboptimal approach. It is anticipated that the general features of the important regions extracted by the teacher model will be summarised and incorporated as part of the student model input. A set of learnable prompts, denoted by $\boldsymbol{P}^E \in \mathbb{R}^{T \times C}$ is introduced, where $T$ represents the length of the prompts and $C$ represents the number of channels, which is consistent with the number of channels in the teacher model features $F^T$. These prompts are sparse and are used to store the characteristic regions predicted by the teacher model. Before training, this is randomly initialised and will be updated dynamically during the distillation process. To prevent the storage of duplicate information, the different prompts are passed through a self-attention layer, making them visible to each other, as shown in the following Eq. 3:

$$\boldsymbol{P}^E = \sum_{m=1}^{M} \boldsymbol{W}_m \left[ Attn(Q = \boldsymbol{P}^E, K = \boldsymbol{P}^E) \cdot \boldsymbol{W}'_m (V = \boldsymbol{P}^E) \right], \tag{3}$$

Subsequently, $\boldsymbol{W}_m$ and $\boldsymbol{W}'_m$ are learnable weights, the query is generated from $\boldsymbol{P}^E$ and the key and value are taken to be the features of the teacher model. The cross-attention mechanism is employed to extract useful feature information from the context predicted by the teacher. The attention mechanism establishes a soft association between each prompt and its corresponding cluster of features,

thus avoiding the hard association artificially introduced for different input images. However, the feature maps from the teacher model contain a significant amount of noisy background information. Interacting with dense feature pixels can lead to a decrease in the efficiency of attention, while also leading to unnecessary computational consumption. In order to obtain keys and values containing useful information in advance, we propose an initialization strategy for the above features. We normalise the channel dimension of the teacher's features and select the top-$N$ pixels as candidates. Furthermore, since object features are often strongly correlated with the category, we combine the feature with a category-aware embedding, which is encoded by the one-hot category vector.

$$\boldsymbol{P}^E = (1 - \beta) \sum_{m=1}^{M} \boldsymbol{W}_m \left[ Attn(Q = \boldsymbol{P}^E, K = Init(F^T)) \cdot \boldsymbol{W}'_m (V = Init(F^T)) \right] + \beta \boldsymbol{P}^E, \quad (4)$$

The update formula for prompt $\boldsymbol{P}^E$ from the teacher model is given by Eq. 4, where $\beta = 0.8$ is the momentum coefficient. It should be noted that the interaction between the prompts and the teacher model is limited to the training phase; in contrast, the prompts trained during the testing phase only participates in subsequent interactions with the student model. Consequently, the strategy of momentum updating gives prompts itself a larger weight, which is beneficial in ensuring that the prompts for the student model is relatively consistent as input during both the training and testing processes. Subsequent to this, the student model takes the generated prompts as input knowledge and also employs the cross-attention mechanism to search for effective information to enhance its representation ability. As illustrated in Eq. 5, we also initialise the features of the student model and utilise them as query. The prompts generates key and value, and the fused query is interpolated and transformed into a residual term.

$$F^S = \sum_{m=1}^{M} \boldsymbol{W}_m \left[ Attn(Q = Init(F^S), K = \boldsymbol{P}^E) \cdot \boldsymbol{W}'_m (V = \boldsymbol{P}^E) \right] + F^S, \quad (5)$$

### 3.2.2 INTERNAL PROMPTS DISTILLATION

Given the substantial disparity in the structure and quantity of parameters between the teacher and student models, as well as the limitations of blindly injecting knowledge from the teacher model, it is also essential to ensure that the student model retains its own effective internal information. To this end, we introduce the learnable prompt as the internal knowledge base of the student model. Since it does not involve cross-model interaction, we adopt the hard association method of dot product. As described in Eq. 6, the prompts $\boldsymbol{P}^I \in \mathbb{R}^{N \times C}$ describes the dependencies of $N$ clusters, and the number of channels $C$ is consistent with the features of the student model. The degree to which pixels are rich in information can be obtained by calculating the similarities between the prompts and the feature map in the spatial dimension.

$$\mathcal{M} = \sigma \left( \boldsymbol{P}^I \times F^S \right), \quad (6)$$

where $\sigma$ denotes Sigmoid function and mask $\mathcal{M} \in \mathbb{R}^{N \times H \times W}$. Similarly, it is necessary to ensure that the different prompts, represented by the matrices $\mathcal{M}_i$, in the prompts focus on different feature templates. As shown in Eq. 7, 8, we utilize the Dice coefficient following MasKD (Huang et al., 2022b) to supervise the learning of prompts.

$$\mathcal{L}_{\text{div}} = \frac{1}{N^2} \sum_{i=1}^{N} \sum_{j=1}^{N} \rho_{\text{dice}} \left( \mathcal{M}_i, \mathcal{M}_j \right), \quad (7)$$

$$\rho_{\text{dice}} \left( \boldsymbol{a}, \boldsymbol{b} \right) = \frac{2 \sum_{i=1}^{M} a_i b_i}{\sum_{j=1}^{M} a_j^2 + \sum_{k=1}^{M} b_k^2}, \quad (8)$$

Dice coefficient $\rho_{\text{dice}}$ is typically employed to assess the resemblance between two vectors. It is regarded as a penalty term for prompts, with the objective of preventing the system from becoming trapped in local optima. Subsequently, the crucial information from the original student features is matched with prompts $\mathcal{M}$, and the enhanced features are obtained as follows:

$F^S = \frac{1}{N} \left( \mathcal{M} \odot F^S \right) + F^S$. Furthermore, an additional ConvLoRA (Aleem et al., 2024) is introduced as a residual term inserted into the student model, which serves to preserve effective features and suppress noise signals through the compression and decompression processes.

# 4 MAIN EXPERIMENTS

## 4.1 DATASETS AND EVALUATION METRIC.

To verify the effectiveness of our method, we conduct extensive experiments on the challenging MS-COCO 2017 dataset (Lin et al., 2014). The COCO dataset contains 80 object classes with 118k images for training and 5k images for testing, respectively. The performance is evaluated by the mean Average Precision (mAP) metric across the IoU threshold from 0.5 to 0.95 over all classes. Specific experimental details and parameter designs are provided in the appendix.

## 4.2 MAIN RESULTS

**Results on different backbones.** Here, we show that our method is effective regardless of the backbone architectures. We utilize GFL (Li et al., 2020) as the detector. Three types of efficient tiny backbones are used by the students, including GhostNet (Han et al., 2020), MobileNetV2 (Sandler et al., 2018) and FasterNet (Chen et al., 2023). The ResNet 101 (He et al., 2016) is used by the teachers. We compared our method with five recent state-of-the-art distillation methods (Yang et al., 2022b;a; Cao et al., 2022; Huang et al., 2022a;b). As shown in Tab. 1, our distillation method surpasses other state-of-the-art distillation methods. All the student detectors gain significant improvements in AP with the knowledge transferred from teacher detectors, *e.g.*, GFL with GhostNet achieves a 3.5% mAP improvement on the COCO dataset. These results indicate the effectiveness and generality of our method across different backbones.

Table 1: Results of the proposed method with different backbones on the COCO dataset. T and S mean the teacher and student detector, respectively.

| Method | schedule | mAP | $AP_{50}$ | $AP_{75}$ | $AP_S$ | $AP_M$ | $AP_L$ |
|---|---|---|---|---|---|---|---|
| GFL-Res101 (T) | 2× | 44.9 | 63.1 | 49.0 | 28.0 | 49.1 | 57.2 |
| GFL-GhostNet (S) | 2× | 32.6 | 49.0 | 35.2 | 18.0 | 35.0 | 43.7 |
| FGD (Yang et al., 2022a) | 2× | 31.4 (-1.2) | 46.7 | 33.7 | 17.2 | 33.1 | 42.7 |
| MGD (Yang et al., 2022b) | 2× | 34.6 (+2.0) | 51.3 | 37.1 | 19.8 | 37.6 | 45.5 |
| PKD (Cao et al., 2022) | 2× | 33.5 (+0.9) | 49.3 | 36.2 | 16.6 | 36.0 | 47.5 |
| DIST (Huang et al., 2022a) | 2× | 33.0 (+0.4) | 49.5 | 35.3 | 17.1 | 35.4 | 45.5 |
| MasKD (Huang et al., 2022b) | 2× | 34.0 (+1.4) | 50.2 | 36.8 | 18.0 | 36.7 | 46.2 |
| Ours | 2× | **36.2 (+3.6)** | **52.8** | **39.2** | **18.4** | **39.1** | **50.1** |
| GFL-MobileNetV2 (S) | 2× | 30.0 | 44.7 | 32.0 | 16.3 | 31.7 | 39.5 |
| FGD (Yang et al., 2022a) | 2× | 33.0 (+3.0) | 48.3 | 35.3 | 18.3 | 35.0 | 44.7 |
| MGD (Yang et al., 2022b) | 2× | 35.1 (+5.1) | 51.1 | 38.0 | 20.0 | 37.3 | 46.9 |
| PKD (Cao et al., 2022) | 2× | 36.5 (+6.5) | 52.8 | 39.5 | 19.9 | 39.7 | 50.0 |
| DIST (Huang et al., 2022a) | 2× | 31.5 (+1.5) | 47.0 | 33.8 | 16.1 | 33.3 | 42.7 |
| MasKD (Huang et al., 2022b) | 2× | 34.6 (+4.6) | 50.3 | 37.5 | 19.4 | 36.8 | 45.8 |
| Ours | 2× | **37.4 (+7.4)** | **53.9** | **40.3** | **20.2** | **41.0** | **51.2** |
| GFL-FasterNet (S) | 2× | 32.5 | 49.2 | 34.5 | 17.5 | 35.3 | 43.2 |
| FGD (Yang et al., 2022a) | 2× | 33.1 (+0.6) | 49.2 | 35.4 | 19.1 | 35.6 | 43.9 |
| MGD (Yang et al., 2022b) | 2× | 34.5 (+2.0) | 51.1 | 37.2 | 19.2 | 37.3 | 46.1 |
| PKD (Cao et al., 2022) | 2× | 36.0 (+3.5) | 52.3 | 38.9 | 18.7 | 39.0 | 49.4 |
| DIST (Huang et al., 2022a) | 2× | 33.1 (+0.6) | 50.1 | 35.3 | 17.2 | 35.8 | 45.2 |
| MasKD (Huang et al., 2022b) | 2× | 35.3 (+2.8) | 51.8 | 37.9 | 18.9 | 37.9 | 47.8 |
| Ours | 2× | **37.7 (+5.2)** | **54.3** | **40.8** | **20.0** | **41.1** | **51.8** |

**Results on different detectors.** Our method can be applied to different detection frameworks easily, so we conduct experiments on three popular detectors, including a two-stage detector (Faster RCNN (Ren et al., 2015)), an anchor-based one-stage detector (RetinaNet (Lin et al., 2017b)) and an anchor-free one-stage detector (RepPoints (Yang et al., 2019)). The same backbone of ResNet 101 and GhostNet is used by the teachers and students respectively. And we compare the results with PKD (Cao et al., 2022), which is another effective and general distillation method. As shown in Tab. 2, our method consistently boosts the performance of all the student-teacher pairs, surpassing the counterpart in all cases.

Table 2: Results of the proposed method with different detection frameworks on the COCO dataset.

| Method | schedule | mAP | $AP_{50}$ | $AP_{75}$ | $AP_S$ | $AP_M$ | $AP_L$ |
|---|---|---|---|---|---|---|---|
| Faster RCNN-Res101 (T) | 2× | 39.8 | 60.1 | 43.3 | 22.5 | 43.6 | 52.8 |
| Faster RCNN-GhostNet (S) | 2× | 28.9 | 47.0 | 30.5 | 16.7 | 30.8 | 38.8 |
| PKD (Cao et al., 2022) | 2× | 30.3 (+1.4) | 47.8 | 32.4 | 15.5 | 33.2 | 42.1 |
| Ours | 2× | **33.0 (+4.1)** | **51.6** | **35.5** | **16.5** | **35.8** | **45.9** |
| RetinaNet-Res101 (T) | 2× | 38.9 | 58.0 | 41.5 | 21.0 | 42.8 | 52.4 |
| RetinaNet-GhostNet (S) | 2× | 29.2 | 47.9 | 30.2 | 15.4 | 31.8 | 39.6 |
| PKD (Cao et al., 2022) | 2× | 29.6 (+0.4) | 46.7 | 31.0 | 16.0 | 32.7 | 40.9 |
| Ours | 2× | **31.9 (+2.7)** | **49.6** | **33.3** | **16.8** | **34.8** | **44.5** |
| RepPoints-Res101 (T) | 2× | 42.9 | 63.8 | 46.5 | 25.1 | 47.1 | 57.0 |
| RepPoints-GhostNet (S) | 2× | 31.6 | 50.3 | 33.3 | 17.6 | 34.1 | 42.6 |
| PKD (Cao et al., 2022) | 2× | 32.6 (+1.0) | 51.0 | 34.2 | 17.5 | 34.9 | 44.8 |
| Ours | 2× | **34.4 (+2.8)** | **53.3** | **36.3** | **18.3** | **36.8** | **47.9** |

Table 3: Results of the proposed method with different heterogeneous detector pairs on the COCO dataset.

| Method | schedule | mAP | $AP_{50}$ | $AP_{75}$ | $AP_S$ | $AP_M$ | $AP_L$ |
|---|---|---|---|---|---|---|---|
| RetinaNet-Res101 (T) | 2× | 38.9 | 58.0 | 41.5 | 21.0 | 42.8 | 52.4 |
| RepPoints-GhostNet (S) | 2× | 31.6 | 50.3 | 33.3 | 17.6 | 34.1 | 42.6 |
| PKD (Cao et al., 2022) | 2× | 31.7 (+0.1) | 50.0 | 33.6 | 16.9 | 34.9 | 42.3 |
| Ours | 2× | **34.1 (+2.5)** | **52.7** | **36.3** | **17.5** | **37.0** | **47.7** |
| Faster RCNN-Res101 (T) | 2× | 39.8 | 60.1 | 43.3 | 22.5 | 43.6 | 52.8 |
| RetinaNet-GhostNet (S) | 2× | 29.2 | 47.9 | 30.2 | 15.4 | 31.8 | 39.6 |
| PKD (Cao et al., 2022) | 2× | 28.5 (-0.7) | 44.8 | 30.0 | 13.8 | 31.6 | 40.7 |
| Ours | 2× | **31.8 (+2.6)** | **49.5** | **33.8** | **16.2** | **34.4** | **45.8** |
| Mask RCNN-Res101 (T) | 2× | 40.8 | 61.0 | 44.5 | 23.0 | 45.0 | 54.1 |
| RetinaNet-GhostNet (S) | 2× | 29.2 | 47.9 | 30.2 | 15.4 | 31.8 | 39.6 |
| PKD (Cao et al., 2022) | 2× | 28.9 (-0.3) | 44.9 | 30.8 | 14.7 | 31.2 | 40.6 |
| Ours | 2× | **31.5 (+2.3)** | **48.8** | **33.1** | **15.9** | **34.3** | **44.1** |
| Mask RCNN-Res101 (T) | 2× | 40.8 | 61.0 | 44.5 | 23.0 | 45.0 | 54.1 |
| GFL-GhostNet (S) | 2× | 32.6 | 49.0 | 35.2 | 18.0 | 35.0 | 43.7 |
| PKD (Cao et al., 2022) | 2× | 31.6 (-1.0) | 46.3 | 34.3 | 15.6 | 33.7 | 45.5 |
| Ours | 2× | **34.6 (+2.0)** | **50.9** | **37.5** | **17.6** | **37.3** | **48.6** |

**Results on heterogeneous detector pairs.** The majority of existing methods have been tailored for homogeneous detector pairs, whereas our approach possesses the versatility to facilitate knowledge transfer across both homogeneous and heterogeneous detector pairs. In this context, we have extended our experimentation to encompass a broader array of detectors, leveraging more sophisticated heterogeneous teacher detectors, as detailed in Tab. 3. Our findings demonstrate that our

method exhibits enhanced adaptability to heterogeneous models and consistently leads to superior performance improvements.

### 4.3 ABLATION STUDIES

**Effects of components in DualPromptKD.** We conducted experiments to demonstrate the impact of each component within our DualPromptKD framework, as detailed in Tab. 4. Our approach encompasses three distinct distillation phases: Feature distillation, Internal prompt distillation, and External prompt distillation. We initially assessed the efficacy of the two novel prompt distillation types in

Table 4: Ablation study of components on GFL ResNet-101 teacher and GFL GhostNet student. FD stands for Feature distillation; EPD stands for External prompt distillation; IPD stands for Internal prompt distillation without ConvLoRA.

| Distillations | AP | $AP_S$ | $AP_M$ | $AP_L$ |
|---|---|---|---|---|
| None | 32.6 | 18.2 | 34.6 | 47.5 |
| FD | 33.5 (**+0.9**) | 16.6 | 36.0 | 47.5 |
| FD + ConvLoRA | 34.2 (**+1.6**) | 17.1 | 35.8 | 48.5 |
| FD + EPD | 34.7 (**+2.1**) | 17.3 | 37.1 | 48.7 |
| FD + IPD + ConvLoRA | 35.6 (**+3.0**) | 18.1 | 37.6 | 49.4 |
| FD + EPD + IPD | 35.8 (**+3.2**) | 17.7 | 38.3 | 50.3 |
| FD + EPD +IPD + ConvLoRA | 36.2 (**+3.6**) | 18.4 | 39.1 | 50.1 |

conjunction with Feature distillation. To better illustrate the roles of different modules, we also conducted ablation experiments on the ConvLoRA. Our findings indicate that each component substantially enhances the student model's performance when utilized individually. Moreover, the synergistic application of all components yields optimal results, suggesting that external and internal prompt distillation each encapsulate distinct aspects of knowledge and are mutually complementary.

**Sensitivity study of the length and dimensions of the prompts.** In the realm of external prompt distillation, we utilize a trainable set of prompts, denoted as $\boldsymbol{P}^E \in \mathbb{R}^{T \times C}$, to capture the distinctive regions highlighted by the teacher model. The prompt dimensions, $\boldsymbol{T}$ and $\boldsymbol{C}$, are pivotal in determining the effectiveness of knowledge transfer. To evaluate their influence, we performed an ablation study, as shown in Fig. 4. Our findings reveal that the model's performance remains stable across various $\boldsymbol{T}$ and $\boldsymbol{C}$ values, with the maximum mAP decrease at 0.3 points from the optimal setup. Further analysis shows that while a larger prompt size can escalate model complexity, the performance gains are marginal. Significantly, the $32 \times 64$ prompt size exemplifies the efficiency of our method, achieving considerable performance improvements with a minor parameter increase. This underscores the fine balance our approach strikes between model complexity and performance enhancement.

**Effectiveness and necessity of the distillation process.** Our DualPromptKD proposes a method that utilizes prompts to enrich the information of the student model, which can also be used without employing distillation. In this section, we validate the effectiveness and necessity of the distillation process, and the experimental results are shown in Fig. 5. The results indicate that while introducing prompts without using distillation does yield some performance improvement, the improvement is quite limited compared to the results obtained with the distillation process. For instance, on MobileNetV2, using only prompts results in a 0.4 mAP improvement, whereas incorporating distillation leads to a 7.4 mAP performance enhancement.

## 5 CONCLUSION

In this paper, we introduce DualPromptKD, an innovative framework for object detection distillation that employs prompts to bolster knowledge transfer between teacher and student models. Our approach integrates three specialized distillation modules: Feature Distillation, Internal Prompt Distillation, and External Prompt Distillation. Utilizing supplementary prompts for knowledge transfer, our method adeptly reduces the performance gap between teacher and student models with a modest overhead. Comprehensive experiments on diverse architectures and detectors affirm the simplicity

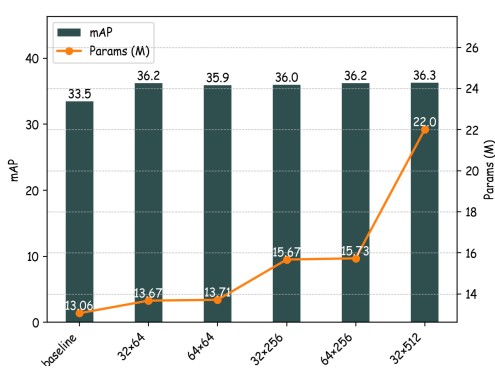

Figure 4: Comparison of model performance and parameter count under different prompt shapes. The horizontal axis represents the length and dimensions of the prompts. The "baseline" denotes the scenario where no prompt is used, i.e., only feature distillation.

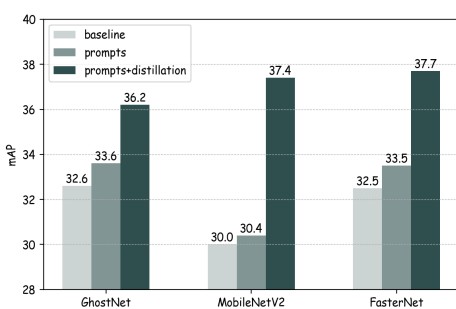

Figure 5: Comparison of student model performance under different conditions. "baseline" represents the original student model, "prompts" indicates the model is applied with prompts without using distillation, "prompts+distillation" signifies that the model incorporates both prompts and distillation. All student models utilize GFL for detectors, and GFL-Res101 is used as the teacher model.

and efficacy of our approach. We envision DualPromptKD as a catalyst for future innovation in the field of knowledge distillation for compact models.

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

# A APPENDIX

## A.1 IMPLEMENTATION DETAILS

We train the student models with a batch size of 16 for 24 epochs (known as a $2\times$ schedule). The initial learning rate is set by 0.01 for one-stage detectors and 0.02 for two-stage detectors. We reduce the learning rate by $0.1\times$ at the 16th and 22nd epochs. We use SGD as the optimizer and set the momentum and weight decay by 0.9 and 0.0001, respectively. All the experiments are conducted on 8 GPUs with mmdetection (Chen et al., 2019) and mmrazor (Contributors, 2021) on PyTorch.

In the Feature Distillation Module, our primary training strategy was inspired by PKD cao2022pkd. For the hyperparameter loss weight, we set it to 100 for both GFL and RepPoint, and to 10 for both RetinaNet and Faster R-CNN. In the External Prompt Distillation module, we employed prompts of size $32\times64$ to convey knowledge, with a momentum coefficient set to 0.8. Within the Internal Prompts Distillation, the dimensionality of the prompts must match the output feature dimensions of each layer of the backbone. For GhostNet, these dimensions are 24, 40, 112, and 160; for MobileNetV2, they are 24, 32, 96, and 1280; and for FasterNet, they are 40, 80, 160, and 320. The length was uniformly set to 8.

## A.2 EXPERIMENTAL RESULTS ON THE SEGMENTATION TASK.

Although our method is designed for object detection tasks, its strong versatility allows it to be applied to different datasets and downstream tasks. In the Tab. 5 below, we present the experimental results on the segmentation task using the Cityscape dataset.

Table 5: The semantic segmentation results on the Cityscape dataset. T and S mean the teacher and student detector, respectively. DualPromptKD represents the use of our method for distillation.

| Model | aACC | mIoU | mAcc |
|---|---|---|---|
| PSPNet-Res101(T) | 96.33 | 79.76 | 86.57 |
| PSPNet-GhostNet(S) | 90.96 | 50.58 | 57.79 |
| DualPromptKD | 92.09 **(+1.64)** | 53.91 **(+3.13)** | 62.76 **(+4.97)** |

## A.3 DISCUSSION

**Limitation.** Our research methodology faces a limitation in enriching the feature representation of the student model using prompts extracted from the teacher model, which introduces noise due to inherent model differences. Furthermore, the limited information obtained through prompts restricts further performance improvement. Future research should address this limitation to achieve more significant gains in performance.

**Broader Impact.** Our research delves into prompt-based distillation techniques for compact models, presenting a versatile and innovative framework. As object detection models gain prevalence in real-world applications, the demand for lightweight networks grows, driving interest in their development. Our meticulously crafted distillation methodology not only enhances the detection capabilities of lightweight networks but also sets the stage for novel approaches and insights in distillation strategies. We anticipate that our contributions will inspire further exploration and innovation in lightweight network optimization and the broader field of knowledge distillation.

