# OpenReview forum: "Transfering Knowledge into Efficient Tiny Models for Object Detection with Dual Prompt Distillation"
_ICLR.cc/2025/Conference — ICLR 2025 Conference Withdrawn Submission_

### Official Review · Reviewer_VrFw · 2024-10-17

**Soundness:** 2
**Presentation:** 3
**Contribution:** 1
**Rating:** 3
**Confidence:** 4

**Summary:**

The paper introduces DualPromptKD, a knowledge distillation framework designed to transfer knowledge from large teacher models to tiny student models for object detection. DualPromptKD employs a dual-prompt distillation strategy, including external prompts that capture teacher model knowledge and internal prompts that enhance the feature extraction capabilities of student model. Experiments on the COCO benchmark demonstrate the effectiveness of DualPromptKD.

**Strengths:**

1. The writing is clear.

2. "Distillation for tiny models" should be a good research question to discuss.

**Weaknesses:**

1. My main concern lies in the unfair comparison with existing methods.

This paper introduces prompts into the student model, which are additional parameters, while the methods compared in this paper do not add these extra parameters to the student model. Of course, the parameter number of prompt is small, as shown in Figure 4, usually about 20 M, but it is important to note that the GhostNet [1] network used in this paper also typically has only 20 M! This means that the student network parameters may have been doubled, so I think the current experimental comparison is very unfair.

[1] Ghostnet: More features from cheap operations, CVPR 2020

2. Lack of effective new insights.

As Lines 48-49 stated, the main difference between this paper's setting and the existing setting is "much smaller and faster models." Therefore, the authors use the GhostNet network as the student network in their experiments. However, facing this new problem, this paper does not offer new insights. The insight in Lines 85-87 was already mentioned in [2]; the logic in Lines 89-90 is a bit strange - if distilling the output features on ResNet-50 does not constrain the consistency of shallow features, wouldn't the consistency of shallow features on small models be better constrained because there are fewer layers? The insight in Lines 91-95 lead to the design of prompts, but as previously stated, the effectiveness of prompts may lie in the introduction of additional parameters, rather than solving specific challenges when "much smaller and faster models" are used as student models. Therefore, I believe this paper lacks effective new insights, especially in-depth analysis of the specific challenges of the new scenario.

[2]  Pkd: General distillation framework for object detectors via pearson correlation coefficient, NeurIPS 2022

Due to the unfairness of the experimental comparison and the lack of effective insights into the distillation task when using tiny models as student networks, I believe this paper does not meet the ICLR standard, and therefore I give a reject.

**Questions:**

Please refer to the Weaknesses.

---

### Official Review · Reviewer_ZwfL · 2024-10-19

**Soundness:** 3
**Presentation:** 3
**Contribution:** 3
**Rating:** 3
**Confidence:** 5

**Summary:**

In this work, the authors foucs on designing an effective knowledge distillation (KD)  method for object detection. Specifically, they first showed that simply applying existing KD methods on (lightweight) object detection is not feasible, and proposed a new method, named DualPromptKD, for this topic. The authors proposed two designs in their model, i.e. external prompt KD and internal prompt KD. Finally, they conducted extensive experiments on the COCO dataset, which demonstrate the effectiveness of the proposed method.

**Strengths:**

- The topic is meaningful. The authors find the existing object detection KD methods are hard to apply on the light-weight setting.

- Good performance. The authors provide extensive experimental results to show the effectiveness of the proposed method.

**Weaknesses:**

- The authors claims they first apply existing KD methods [1,2,3,4]  on the object detection task [5,6] but find limited improvement (e.g. line 14-17, line 50-53), which is the main motivation of this work. However, these baseline KD methods are somewhat outdated are not the SOTAs (line 52), and more recent works should be investigated. Also, the experimental results should compare the most recent KD methods.

- This work mainly focus on KD for lightweight object detection. With this scope, I beleive the latency of the detectors should be listed. Besides, the authors mainly conducted the experiments with light-weight backbones, which is okay for this paper, but I still expect two further investigation: 1. Can the proposed KD method also work for the general backbones?  2. More experiments with efficient object detectors (such as EfficientDet, YOLOv4/7 etc.) should be conducted.


[1] Focal and global knowledge distillation for detectors, CVPR'22

[2] Pkd: General distillation framework for object detectors via pearson correlation coefficient, NeurIPS'22

[3] Knowledge distillation from a stronger teacher, arXiv'22

[4] Masked distillation with receptive tokens, arXiv'22

[5] Generalized focal loss: Learning qualified and distributed bounding boxes for dense object detection, NeurIPS'20

[6] Ghostnet: More features from cheap operations, CVPR'20

**Questions:**

Overall, I think that investigating KD methods for lightweight object detection is meaningful. However, the current version can not fully support the claims. I expect more experiments in the rebuttal phase, including:

- experimental comparions with most recent works to support the claims and demonste the superiority of this work;

- more experiments with efficient detection pipelines, which can strengthen the core contribution of this work;

- more experiments with general backbones to show the wide applications, which is a bonus of this work.

I consider to improve my rating if these concerns can be addressed.

---

### Official Review · Reviewer_RZW6 · 2024-10-29

**Soundness:** 2
**Presentation:** 2
**Contribution:** 2
**Rating:** 6
**Confidence:** 4

**Summary:**

This paper proposes a simple prompt-based object detection distillation framework, termed DualPromptKD, which aims to improve knowledge transfer efficiency from both teacher and student perspectives. The author enables the student model to fully leverage proficient teacher knowledge by distilling teacher representations into compact external prompts. In terms of the limited learning ability of the student model, the author introduces lightweight internal prompts tailored to bolster the feature imitation capability for the target model. Comprehensive experiments on MS COCO dataset demonstrate the effectiveness of proposed DualPromptKD.

**Strengths:**

- This paper attempt to use prompt learning methods in object detection KD. The expeirments are sufficient to demonstrate the effectiveness.

**Weaknesses:**

- Additional prompt learning modules and ConvLoRA bring extra parameters to the student, which goes against the purpose of compressing model of KD.

**Questions:**

- In line 241~242, the author states that **the gradient of KD signals is prone to disappear in shallow stages, making it difficult to optimise effectively**. it is not a consensus as for me. It's better to cite some previous works or do some experiments to prove this statement.
- What does $M$ denote in formula (3)?
- How model select the top-N pixels in $Init$?
- The inputs and outputs of most formulas in this paper use the same symbol, which makes it harder to understand. It's better to use different symbols to distinguish output from input.

---

### Official Review · Reviewer_qW7a · 2024-11-03

**Soundness:** 2
**Presentation:** 2
**Contribution:** 2
**Rating:** 3
**Confidence:** 4

**Summary:**

This paper presents a knowledge distillation method to transfer knowledge from super large teacher object detectors to tiny student detectors. The proposed method incorporates three key distillation components: feature distillation, alongside external and internal prompt distillation mechanisms. The authors make use of the learnable prompts as bridges to mitigate the knowledge gap between teacher and student architectures by transfering the crucial information via the cross-attention modules. To validate their approach, experiments were conducted across various teacher-student detector pairs.

**Strengths:**

+ This work attempts to explore efficient knowledge transfer from super large teachers to tiny students for object detection, which is interesting to the community.

+ The methed that integrates visual prompts with the knowledge distillation framework, bridging the gap between teacher and student models is somehow novel.

+ The experiments across diverse knowledge distillation settings show some promising results.

**Weaknesses:**

However, there are several weaknesses as follows:

1. The presentation of the method section is not well-organized. The authors fail to provide a formulation of the total loss functions employed during distillation, and the optimization process of learnable components remains unclear. Moreover, the mathematical framework fails to establish clear connections between individual equations and their roles in the final losses. It would be better to show comprehensive formulations linking the learnable modules to their corresponding loss terms.

2. The Figure 2 is notably limited to comparing feature maps between only the largest teacher and smallest student models, while omitting features from models of intermediate scales. Thus it is hard to understand the crucial relationship between model size differences and feature representation gaps.

3. Tables 1, 2, 3 are missing essential metrics, such as the number of parameters and inference latency. The proposed KD method will introduce additional parameters and computational overhead during inference compared to other KD methods, which is potentially not fair. A thorough analysis of the additional computational burden and its impact on inference time should be discussed.

4. The experiments are limited to the CNN-based detectors, which raises questions about the method's generalizability to transformer-based detectors.

5. Moreover, the "tiny" students used in this paper are relatively large compared to some truly lightweight detectors (e.g., Tiny-YOLO, SSDLite, EfficientDet, TinyDet, etc.), leaving the method's effectiveness on extremely compressed models unexplored.

**Questions:**

Please see the weaknesses.

---

### Note · Authors · 2024-11-13

I have read and agree with the venue's withdrawal policy on behalf of myself and my co-authors.